# Oligomeric and Fibrillar Species of Aβ42 Diversely Affect Human Neural Stem Cells

**DOI:** 10.3390/ijms22179537

**Published:** 2021-09-02

**Authors:** Adela Bernabeu-Zornoza, Raquel Coronel, Charlotte Palmer, Victoria López-Alonso, Isabel Liste

**Affiliations:** 1Unidad de Regeneración Neural, Unidad Funcional de Investigación de Enfermedades Crónicas (UFIEC), Instituto de Salud Carlos III (ISCIII), Majadahonda, 28220 Madrid, Spain; adela.bernabeu@gmail.com (A.B.-Z.); raquelicoronel@gmail.com (R.C.); cpalmer248@gmail.com (C.P.); 2Unidad de Biología Computacional, Unidad Funcional de Investigación de Enfermedades Crónicas (UFIEC), Instituto de Salud Carlos III (ISCIII), Majadahonda, 28220 Madrid, Spain; victorialopez@isciii.es

**Keywords:** Alzheimer’s, Aβ peptide, Aβ42, oligomers, fibrils, cell differentiation, cell death, human NSCs

## Abstract

Amyloid-β 42 peptide (Aβ_1-42_ (Aβ42)) is well-known for its involvement in the development of Alzheimer’s disease (AD). Aβ42 accumulates and aggregates in fibers that precipitate in the form of plaques in the brain causing toxicity; however, like other forms of Aβ peptide, the role of these peptides remains unclear. Here we analyze and compare the effects of oligomeric and fibrillary Aβ42 peptide on the biology (cell death, proliferative rate, and cell fate specification) of differentiating human neural stem cells (hNS1 cell line). By using the hNS1 cells we found that, at high concentrations, oligomeric and fibrillary Aβ42 peptides provoke apoptotic cellular death and damage of DNA in these cells, but Aβ42 fibrils have the strongest effect. The data also show that both oligomeric and fibrillar Aβ42 peptides decrease cellular proliferation but Aβ42 oligomers have the greatest effect. Finally, both, oligomers and fibrils favor gliogenesis and neurogenesis in hNS1 cells, although, in this case, the effect is more prominent in oligomers. All together the findings of this study may contribute to a better understanding of the molecular mechanisms involved in the pathology of AD and to the development of human neural stem cell-based therapies for AD treatment.

## 1. Introduction

The characteristic dementia of Alzheimer’s disease (AD) refers to a cognitive and functional decline associated with age together with a particular neuropathology. After years of study, it is known that changes in the cleavage of the amyloid precursor protein (APP) and the production of fragments of amyloid beta (Aβ), together with the aggregation of the hyperphosphorylated tau protein, are the causes of the disease, whereby they fuse to cause a reduction in synaptic strength, synaptic loss, and neurodegeneration [1].

Aβ is a 38- to 43-amino-acid-long peptide generated by APP cleavage [2] and its overproduction generates aggregates (oligomers and fibrils) that play an important role in the development of AD. Aβ was isolated from AD brains and its aggregation into Aβ plaques was first recognized in 1984 [3]. It is also known that the loss of synapses is the structural change that correlates best with the cognitive decline in AD patients and this neuronal dysfunction is induced by the aggregates of Aβ, which are considered the real toxic species in AD [4]. In addition to Aβ fibrils that give rise plaque formation, it is well-known that soluble Aβ oligomers can spread throughout the brain and their distribution is heterogeneous, so it is believed that Aβ oligomers also play an important role in AD. However, there is confusion and controversy about what types and sizes of oligomers have disease-relevant activity [5,6,7].

The function of Aβ oligomers is controversial and the existing results differ greatly between studies [8,9,10]. It has been shown that human neural stem cells (NSCs) exposed to oligomeric Aβ peptides exhibit reduced proliferation and a differentiation commitment to a glial fate, without affecting neuronal fates [11]. This is consistent with a marked effect of Aβ oligomers on NSCs. However, another group found that in NSCs from the rat hippocampus, neurogenesis was induced by oligomeric Aβ42 peptide [11,12]. Furthermore, Heo’s group [13] showed that, at low concentrations, oligomeric Aβ peptide significantly increases the number of proliferating mouse neural precursor cells (NPCs).

Additionally, several groups have observed that Aβ, in its oligomeric form and at low concentrations, does not appear to influence apoptosis [12,14], suggesting that low concentrations of Aβ peptide do not seem to be involved in the process of cell death, but rather might have a neuroprotective effect by promoting the differentiation of NSCs/NPCs. However, upon an increase in its concentration, oligomeric Aβ peptide begins to show its characteristic cytotoxic effects.

The toxic effects of oligomeric Aβ vary depending on whether the peptide is found in a soluble or insoluble form [11,15,16]. Some authors have shown that soluble forms of Aβ oligomers exhibit strong neurotoxic effects and an increase in soluble oligomeric Aβ levels could be a potential cause of AD [17,18]. This suggests that soluble oligomeric Aβ forms are the most toxic species, particularly to neurons, and they are believed to be involved in problems of memory, dementia, and synaptic depletion [11,13,19,20,21].

Studies of the toxic effects of Aβ oligomers have strongly been promoted in AD research because they provide potential explanations for one of the pathological causes of the disease [7,22]. Furthermore, Aβ oligomers could be capable of mediating toxicity by altering neurotransmission, leading to cell death at a distance, which has been shown both in vitro and in vivo [18]. Although the involvement of the Aβ peptide in its oligomeric form has been described in the context of AD, the cause of the disease is still unknown [4,11].

Despite the statement above, many studies claim that Aβ peptide aggregation into fibrils is the key to the pathogenesis of AD. Aβ fibrils are larger than oligomers, insoluble, and assemble into amyloid plaques forming histological lesions that are characteristic of AD [5]. In vitro studies suggest that fibrillary Aβ peptides induce neurotoxicity, mediated by their interaction with neuronal membrane proteins, including APP [23], and that fibrillary Aβ deposits lead to synaptic abnormalities by breaking neuronal branches. Furthermore, it is widely documented that fibrillary Aβ aggregates, which assemble into amyloid plaques in the AD brain, activate microglia, increasing the expression of inflammatory cytokines, which has severe neurodegenerative effects [7]. Some authors have also observed that fibrillary Aβ peptides inhibit neurogenesis by promoting the differentiation of NSCs into glial cells [24].

The accumulation of Aβ peptide in fibril form and the eventual formation of amyloid plaques are also believed to be involved in the hyperphosphorylation of Tau protein and the formation of intracellular tangles [6]. It is believed that Tau protein becomes hyperphosphorylated in response to changes in kinase/phosphatase activity mediated by Aβ aggregation, leading to the formation of neurofibrillary tangles, further contributing to the pathogenesis of AD [1]. That is another reason why fibrillary Aβ peptides are considered so neurotoxic.

However, the idea that amyloid pathology is caused by insoluble fibrillary Aβ or soluble oligomeric Aβ has come under doubt [25], and today an open debate exists about which form is the most toxic. For this reason, more studies are necessary to clarify the functions of these two aggregation states of Aβ peptide.

Studies in human NSCs have provided a useful tool to progress clinically in stem cell-based therapies for several neurodegenerative disorders and have facilitated a better understanding of human brain development and the molecular pathology associated with neurodegeneration. NSCs are multipotent stem cells with the potential to self-renew and to differentiate into the main cellular phenotypes (neurons, astrocytes, oligodendrocytes) of the central nervous system (CNS) [26].

The objective of the present work consists in analyzing and compares the effects of both aggregation forms (oligomeric and fibrils) of Aβ42 peptide on the biology of human NSCs (cell survival/death, cellular proliferation, and phenotypic specification). We tested different doses of oligomer and fibril Aβ42 peptide in an hNS1 cell line undergoing differentiation. These results together with those previously obtained by our group provide evidence that Aβ peptides affect the properties and biology of human NSCs and therefore could contribute to the advancement of therapies based in the use of human NSCs for the treatment of AD.

## 2. Results

The toxic effects of Aβ oligomers and fibrils are well-known due to their involvement in AD, but mainly at high concentrations. However, like other forms of Aβ peptide, the role of these peptides in the biology of human NSCs remains unclear. Here, we analyzed the effects of Aβ42 oligomers (Figure 1A) and fibrils (Figure 1C) on cell death, proliferation, and cell fate specification during the differentiation of hNS1 cells. To confirm that peptides were in the desired oligomeric (Figure 1B) and fibrillary (Figure 1D) forms, we performed Western blot analyses with the different experimental concentrations (0.5, 1, and 5 µM) and the control group.

### 2.1. Effects of Oligomeric and Fibrillary Aβ42 Peptide on Apoptotic Cell Death of Differentiating hNS1 Cells

In order to deepen our analysis of the effect of oligomeric and fibrillary Aβ42 peptide on cell death, we studied the abundance of activated caspase 3 and analyzed the presence of fragmented/pyknotic nuclei and the presence of γH2AFX.

Our first analysis showed no differences between control groups and groups treated with the lowest concentrations of oligomeric Aβ42 (0.5 and 1 µM) when comparing cell morphology in phase contrast images. However, when hNS1 cells were treated with the highest concentration (5 µM), altered morphology throughout the culture was observed and cells appeared damaged and in lower densities (Figure 2A, upper panels). Similar results were observed for Aβ42 fibrils where no differences between the control groups and the group treated with the lowest concentration of fibrillary Aβ42 (0.5 µM) were detected (Figure 2E, upper panels). However, when hNS1 cells were treated with 1 µM and 5 µM of the fibrillary Aβ42 peptide, cells appeared more damaged and formed clusters (Figure 2E, upper panels). Together, these results suggest that only the highest concentration of oligomeric Aβ42 peptide used in this study (5 µM) induced cell death in hNS1 cells as opposed to lower concentrations, while all concentrations of fibrillary Aβ42 peptide provoked an increase of cell death of hNS1 cells.

To further study the effect on cell death, we analyzed the percentage of positive cells for activated caspase 3 (Figure 2A, middle panels, and Figure 2B). Regarding oligomeric Aβ42 peptide, we observed that in the control groups the percentage of apoptotic cells was low (untreated: 1 ± 0.2%, vehicle group: 1.2 ± 0.2%) and in the group treated with the lowest concentration, the percentage of apoptotic cells did not differ significantly from that found in control groups (0.5 µM: 1.5 ± 0.5%; *p* > 0.05; *n* = 3) (Figure 2B). However, when cells were treated with the oligomeric Aβ42 peptide at concentrations of 1 µM and 5 µM, significant differences in the percentage caspase 3^+^ cells were observed (2 ± 0.3% and 4 ± 0.4%; *** *p* < 0.001; *n* = 3, respectively) (Figure 2B). On the other hand, analyzing the percentage of activated caspase 3^+^ cells for Aβ42 fibrils, we observed that in control groups the percentage of apoptotic cells was also reduced (untreated: 2.17 ± 0.7%, vehicle group: 2.12 ± 0.9%) (Figure 2E, middle panels, and Figure 2F). However, in the group treated with the lowest concentration, the percentage of apoptotic cells showed a significant increase (0.5 µM: 10.2 ± 1.75%; *** *p* < 0.001; *n* = 3) (Figure 2F) and continued rising at 1 µM and 5 µM of fibrillary Aβ42 peptide, where we saw a significant increase in the percentage of caspase 3^+^ (10.7 ± 1.3% and 16 ± 1.6%; *** *p* < 0.001; *n* = 3, respectively) (Figure 2F). These results strongly suggest that both oligomeric and fibrillary Aβ42 peptides appear to be involved in the process of apoptotic cell death in a dose-dependent manner, but the effect of fibrillary Aβ42 peptide is stronger than oligomeric Aβ42 peptide (Figure 2I,J).

When analyzing the percentage of pyknotic nuclei, we saw a significant increase after treatment with oligomeric Aβ42 at 1 µM, but not at 0.5 µM (Figure 2A, bottom panels, and Figure 2C). Only 3.08 ± 0.4% of cells in the vehicle group presented pyknotic nuclei, 5.12 ± 1.02% at 0.5 µM (*p* > 0.05; *n* = 3) and 7.44 ± 0.3% at 1 µM (* *p* < 0.05; *n* = 3) (Figure 2C). The most notable effect observed was that 9.3 ± 1.02% (** *p* < 0.01; *n* = 3) of the total cells at 5 µM were positive for fragmented nuclei (Figure 2C), confirming the toxicity of the highest dose. Furthermore, we saw a significant increase after treatment with fibrillary Aβ42 at all concentrations tested (Figure 2E, bottom panels, and 2G). Only 1.9 ± 0.7% of cells in the vehicle group presented pyknotic nuclei, which rose to 4.4 ± 0.8% at 0.5 µM (* *p* < 0.05; *n* = 3) and 4.35 ± 0.6% at 1 µM (* *p* < 0.05; *n* = 3) (Figure 2G). The most remarkable effect observed was that 6.4 ± 0.7% (** *p* < 0.01; *n* = 3) of total cells at 5 µM were positive for fragmented nuclei (Figure 2G), confirming what we observed for caspase 3 immunoreactivity (Figure 2F).

We finally analyzed the abundance of γH2AFX. As shown in Figure 2D, treatment with oligomeric Aβ42 induced a slight increase in γH2AFX. However, treatment with fibrillary Aβ42 provoked a clearly dose-dependent increase in γH2AFX, mainly at 5 µM, indicating the presence of DNA damage and confirming the data observed for cell death (Figure 2H).

### 2.2. Effects of Oligomeric and Fibrillary Aβ42 Peptide on Proliferation of Differentiating hNS1 Cells

To study the effects of proliferation after treatment of oligomeric and fibrillary Aβ42 peptide, the cultures were analyzed using the cell cycle marker Ki67 and the proliferation marker BrdU (Figure 3A,F).

Regarding oligomeric Aβ42 peptide, the cell count revealed that the percentage of Ki67 positive cells in the control groups represented 20 ± 0.7% for the untreated group and 19 ± 1.1% in vehicle group (Figure 3A,C). These results were similar at 0.5 µM, where 20 ± 0.9% (*p* > 0.05; *n* = 3) of cells were Ki67^+^ (Figure 3C). However, we saw an important decline in Ki67^+^ cells at 1 µM in which 16 ± 0.7% (*p* > 0.05; *n* = 3) of cells, and at 5 µM in which 15 ± 0.8% (* *p* < 0.05; *n* = 3) of cells, were positive for the cell cycle marker (Figure 3C). The results were confirmed by qRT-PCR where we observed a decline in *MKi67* mRNA levels at all concentrations tested (Figure 3D). In the case of BrdU (Figure 3B,E), our results confirmed that 10 ± 1.5% of control cells incorporated BrdU. Contrary to the results observed for Ki67, after treatment with oligomeric Aβ42, no significant differences in the number of BrdU^+^ cells were observed at the different doses tested (13 ± 1.8% at 0.5 µM, 12 ± 1.6% at 1 µM and 10 ± 1.6% at 5 µM; *p* > 0.05; *n* = 3) compared to control groups (Figure 3E). All these results suggest that Aβ42 peptide in its oligomeric form appears to have effects on the proliferation of hNS1 cells, essentially by inducing cell cycle exit, without affecting mitosis.

Moreover, results for fibrillary Aβ42 peptide showed that the percentage of Ki67^+^ cells in the control groups represented 17.8 ± 1.4% for the untreated group and 18 ± 1.5% in the vehicle group (Figure 3F,H). These results were similar for the groups treated with 0.5 µM (15.6 ± 1.6%; *p* > 0.05; *n* = 3), 1 µM (15.4 ± 1.9%; *p* > 0.05; *n* = 3), and 5 µM (15.2 ± 1.7%; *p* > 0.05; *n* = 3) (Figure 3H). The results were confirmed by qRT-PCR (Figure 3I). Finally, after a 2 h pulse of BrdU (Figure 3G), our results confirmed that 9 ± 1.5% of control cells incorporated BrdU. After treatment with fibrillary Aβ42, no significant differences in the number of BrdU^+^ cells were observed at the different concentrations tested: 7.2 ± 1.7% at 0.5 µM, 6 ± 1.8% at 1 µM, and 7 ± 2% at 5 µM (*p* > 0.05; *n* = 3) (Figure 3J). Contrary to the effects observed for oligomeric Aβ42 peptide, all these results suggest that Aβ42 peptide in its fibrillary form does not appear to have significant effects on the proliferation of hNS1 cells (Figure 3K,L).

### 2.3. Role of Oligomeric and Fibrillary Aβ42 Peptide in Cell Fate Specification of hNS1 Cells

To study the effects of oligomeric and fibrillary Aβ42 peptide on the cell fate specification of hNS1 cells, we studied β-III-tubulin as a neuronal marker and GFAP as a glial marker (Figure 4A,D and Figure 5A,D).

The results obtained demonstrated that there were significant differences in all groups treated with oligomeric Aβ42 peptide for the neuronal marker β-III-tubulin, which showed an increase in percentage in all groups tested (0.5 µM: 30 ± 0.9%, 1 µM: 31 ± 0.8%, and 5 µM: 40 ± 1.08%; ** *p* < 0.01; *** *p* < 0.001; *n* = 3) with respect to control groups (untreated: 20 ± 0.9%, vehicle: 22 ± 1.2%) (Figure 4A,B). These data were confirmed by qRT-PCR (Figure 4C). 

Furthermore, results obtained after the analysis of cell fate specification revealed that there were also significant differences between the groups treated with the fibrillary Aβ42 peptide and the control groups for β-III-tubulin marker. We observed a significant increase in the number β-III-tubulin^+^ cells after treatment with 1 µM and 5 µM (23.7 ± 0.7%; *p* > 0.05 at 0.5 µM, 25 ± 0.7%; * *p* < 0.05 at 1 µM and 29.8 ± 0.8%; ** *p* < 0.01 at 5 µM; *n* = 3) compared to control groups (untreated: 19.7 ± 1.6%, vehicle: 19.5 ± 1.6%) (Figure 4D,E), and these results were confirmed by qRT-PCR (Figure 4F). According to these results, oligomeric and fibrillary Aβ42 peptides increase neurogenesis of hNS1 cells, with a greater effect being observed in oligomers under our experimental conditions (Figure 4G).

Regarding glial cell fate determination, we observed significant differences in GFAP^+^ cells after treatment with oligomeric Aβ42 peptide (untreated: 6 ± 1.9%, vehicle: 7 ± 1.1%) (Figure 5A–C). At 0.5 µM, an increase in the amount of GFAP^+^ cells (15 ± 1.3%; ** *p* < 0.01; *n* = 3) was observed (Figure 5B). The most significant effect appeared when hNS1 cells were treated with a concentration of 1 µM of oligomeric Aβ42 peptide, where the percentage of positive cells was 17 ± 0.43% (*** *p* < 0.001; *n* = 3). At 5 µM, the percentage dropped to 13 ± 0.86% (* *p* < 0.05; *n* = 3) (Figure 5B).

Regarding fibrillary Aβ42 peptide, we also observed significant differences in GFAP^+^ cells in treated groups compared to control groups (untreated: 10.4 ± 1.9%, vehicle: 9.6 ± 1.6%) (Figure 5D,E). At 0.5 µM, an increase in the number of GFAP^+^ cells (14.3 ± 0.9%; * *p* < 0.05; *n* = 3) was observed (Figure 5E). The most significant effect appeared when hNS1 cells were treated with 1 µM of fibrillary Aβ42 peptide (16.4 ± 1.5%; ** *p* < 0.01; *n* = 3). This then declined at 5 µM (12.6 ± 1.4%; *p* > 0.05; *n* = 3) (Figure 5E). These results were confirmed by qRT-PCR experiment (Figure 5F). This suggests that both oligomeric and fibrillary Aβ42 peptides have positive effects on gliogenesis in a dose-dependent manner, especially at 0.5 µM and 1 µM (Figure 5G).

## 3. Discussion

AD is one of the most prevalent diseases in our society; however, there is still no cure for this disorder. There are some potentially therapeutic strategies based on NSCs; however, the percentages of success are not significant. The application of human NSCs in stem cell therapy for neurodegenerative disorders depends on the ability of transplanted NSCs to survive in affected brains and to proliferate, migrate, and differentiate into different functional neuronal lineages [27,28,29]. For this reason, greater effectiveness is important when it comes to finding possible therapies, and a better understanding of pathological causes, in this case of AD, has become necessary, including the physiological and pathological effects of Aβ peptides.

In an attempt to find possible therapies for this disease, much attention has been focused on understanding the pathophysiological causes of AD, including the toxic effects of Aβ peptides in promoting synaptic dysfunction, neuronal death, and the deterioration of the development of neuronal progenitor cells [29,30,31].

In this work, we analyzed the effects of Aβ42 peptide in its oligomeric and fibrillary forms during 4.5 days of differentiation of hNS1 cells (human NSCs line). We studied their implications for the cell death, cell proliferation, and cell fate specification of hNS1 cells after Aβ treatment.

### 3.1. Effects of Oligomeric Aβ42 Peptide

Regarding the studies for cell death, the results obtained from activated caspase 3, pyknotic nuclei, and the expression of γH2AFX suggest that oligomeric Aβ42 peptide is harmful for cell survival at high concentrations. Our results are similar to those obtained by Lee et al. [11], where the presence of Aβ42 oligomers increased the number of TUNEL+ cells, indicating apoptosis. Although treatment with Aβ42 peptide has significant effects on apoptosis compared to control groups, at low concentrations oligomeric Aβ42 peptide does not seem to be harmful [13], since a significant effect on differentiation was also observed. Therefore, it can be suggested that low concentrations of Aβ oligomers are related to the determination of the cellular fate of hNS1 cells.

When changes in the proliferation rate were analyzed, the results obtained for the cell cycle protein Ki67 and proliferation marker BrdU suggested that the oligomeric state of Aβ42 peptide appears to be involved in causing cell cycle alteration. We observed a decrease in proliferation, mainly at high concentrations, perhaps due to the increase in cell death. These results are in line with the results obtained by López-Toledano and Shelanski [12] and Lee et al. [11], where no significant differences in cell proliferation were observed when comparing Aβ treatments and controls. Similar to the results obtained by He et al. [32], proliferation is not altered when treating neural precursors with low doses of Aβ oligomers.

Despite what was mentioned before, the function of oligomeric Aβ peptide in cell fate specification is very controversial and differs in various studies already published [23,31,32]. It has been shown that human NSC exposure to oligomeric Aβ peptides, mainly Aβ42, decreased the proliferative potential of these cells, stimulating their differentiation into a glial cell fate without affecting neuronal fates [10]. However, another group found that in NSCs from the rat hippocampus, neurogenesis was induced by oligomeric Aβ42 and this activity was associated with Aβ oligomers and not with fibrils [12].

Altogether, this suggests that the effects of these peptides may mostly affect cells that are in a post-mitotic state, being involved in cell fate, differentiation, and death of mature cells. In view of these results and the corresponding bibliography, it can be suggested that hNS1 cells subjected to low concentrations of oligomeric Aβ peptides undergo cell cycle exit in order to enter a senescence process through several possible routes, such as the routes involving MAPK, GSK3β, or telomeric shortening [10,30]. According to the results obtained by Lee et al. [11], oligomeric Aβ42 peptide does not promote cell proliferation, which corresponds with the results obtained by other groups [12,32]

In view of the great contradictions that exist in the field, we tried to understand the effect of oligomeric Aβ peptides on the differentiation of hNS1 cells. According to the results obtained by López-Toledano and Shelanski [12] and Lee et al. [11], the oligomeric form of Aβ42 peptide is involved in neurogenesis, with 1 μM being the optimal concentration at which this effect is observed. In our case, it can be observed that this effect appears at even lower concentrations, since when hNS1 cells were treated with 0.5 μM of oligomeric Aβ42, a clear effect in neurogenesis could be found, very similar to that observed in the other concentrations studied. These results correspond to data obtained in studies of monomers by other groups [13,26,33], and partially with the results of Lee et al. [11], where it was explained that there is differentiation towards neural destiny. Regarding cell differentiation, it has been observed that the process at neurogenesis may be due to the action of Aβ peptide on NSCs [5,12].

When hNS1 cells were treated with the oligomeric Aβ42 peptide, they were also able to differentiate into glial cells. These results match with those of other studies conducted on nonhuman primates [34], where oligomeric forms of Aβ peptides were found to activate the glia, causing AD. These results were very similar to those observed when treating the peptides in their monomeric form [35,36]. In addition, the gliogenesis process was mostly observed at low concentrations (0.5 and 1 μM), which corresponds to the data obtained by Lee et al. [11], where it was shown that glial differentiation was increased when treating cells with 1 μM of oligomeric Aβ42 peptide. However, when the concentration rose (5 μM), a decrease in the number of GFAP+ cells could be observed, which may have been partly due to the increase in cell death. 

### 3.2. Effects of Fibrillary Aβ42 Peptide

Finally, it is already known that Aβ peptide in its fibrillary state is the predominant form that accumulates in AD brains, forming amyloid plaques [37]. Furthermore, in vitro experiments have demonstrated that Aβ42 aggregates into amyloid plaques much more rapidly than Aβ40, confirming that Aβ42 is the main isoform present in these plaques [38,39]. To study the possible physiological functions of fibrillary Aβ42 peptides during differentiation of human NSCs, we analyzed the effects of different concentrations of this Aβ peptide on cell death, cell fate specification, and cell proliferation of hNS1 cells.

Aβ neurotoxicity has been shown to correlate with the presence of fibrils. Our results showed that all concentrations of fibrillary Aβ42 peptide tested induced programmed cell death of hNS1 cells, as confirmed by activated caspase 3, pyknotic nuclei quantification, and γH2AFX expression. These results are in concordance with those obtained by other groups. Some authors have observed that human NPCs treated with fibrillary Aβ42 present an increase in the percentage of caspase 3+ cells [40], while other authors have observed a decrease in the number of total cells (mainly neurons) after treatment with 1 μM of fibrillary Aβ42 peptide [13,41].

Numerous studies have also shown the implication of Aβ peptide in NSC proliferation. This effect depends on the state in which the peptide is found, the type of cells, and the time of treatment. In our results, we observed, from Ki67 expression and BrdU incorporation, that fibrillary Aβ42 peptide has no significant effect on proliferation of hNS1 cells. These results are contrary to those obtained by Porayette et al.’s group [40], who observed an increase in the number of human NPCs after treatment with fibrillary Aβ42. However, although the doses tested were the same, these discrepancies could have been due to the different times of treatment registered in the experiment (10 days vs. 4.5 days).

Though the toxic effect of fibrillary Aβ peptides has been widely documented, our results provide evidence that fibrillary Aβ42 could promote neural differentiation of human NSCs. We found that treatment with fibrillary Aβ42 peptide produces a significant increase in neurogenesis and that this increment is concentration-dependent. This is contrary to other results, which showed that treatment with low concentrations of fibrillary Aβ42 does not affect the neuronal differentiation of human NPCs [13].

On the other hand, we also observed from GFAP expression that fibrillary Aβ42 peptide produces a significant increase in gliogenesis. Similar results obtained by Malmten et al.’s group [24] showed that treatment with 1 μM of fibrillary Aβ42 provoked an increase in cells that differentiated towards a glial phenotype.

To conclude, our results show that Aβ42 oligomers decrease the proliferation of hNS1 cells, while in the case of fibrils, proliferation is not affected. All forms of aggregation seemed to present an increase in the differentiation of cells towards a glial destination. When the cells were treated with oligomers or fibrils, an increase in the generation of neurons was observed. Finally, with regard to toxicity, although all forms (oligomers and fibrils) showed increased cell death for hNS1 cells, the fibrillary form was the one with the highest levels of apoptosis at the same doses tested (Figure 6).

It is widely known that Aβ mediates neuronal cell death, synaptic dysfunction, and neurodegeneration. It has been suggested that, depending on the size of the oligomers, these species are capable of binding to some receptors, triggering different responses by activating different signal pathways in NSCs. Knowing the molecular mechanisms that regulate the proliferation and differentiation of NSCs in a neurodegenerative environment can provide valuable information for potential therapies based on the use of stem cells. Taken together, it can help us understand the cellular and molecular processes that occur in the brains of Alzheimer’s patients and which point out treatment that could be more effective. A major limitation of these types of studies is the lack of models, both in vitro and in vivo, that perfectly mimic brains affected by Alzheimer’s. However, our cell system may be a useful tool to study the physiological context of a brain with AD and help clinical progress in stem cell-based therapies to treat this disease.

## 4. Materials and Methods

### 4.1. Ethics Statement

hNS1 cells were obtained from human tissues donated for research after written informed consent, in accordance with the European Union directives and the declaration of Helsinki and in agreement with the ethical guidelines of the Network of European CNS Transplantation and Restoration (NECTAR) and Spanish Biomedical Research Law (July 2007). Ethics statements about the human fetal origin of the cells used here can be found in the original reports describing the cell line [28,42,43,44]. The study was approved by the Ethics Committee of the Instituto de Salud Carlos III (approval number PI93-2020, 1 December 2020).

### 4.2. Cell Cultures

Isolation and immortalization of hNS1 cells has been described previously [28,42,43,44]. hNS1 cell culture conditions were based on a chemically defined HSC medium (Dulbecco’s Modified Eagle Medium (DMEM): F12 (1:1) with GlutaMAX-I medium (Gibco, Langley, OK, USA) containing 1% AlbuMAX (Gibco), 50 mM HEPES (Gibco), 0.6% d-glucose (Merck, Darmstadt, Germany), 1% N-2 supplement (Gibco), 1% non-essential amino acids mixture (NEAA; Gibco), and 1% penicillin–streptomycin (P/S; Lonza, Switzerland). For experiments, cells were seeded at 15,000 cell/cm^2^ on poly-l-lysine (10 µg/mL; Sigma, Darmstadt, Germany)-coated plastic dishes. Cells were grown in HSC medium supplemented with 20 ng/mL of epidermal growth factor (EGF; PeproTech, London, UK) and 20 ng/mL of fibroblast growth factor 2 (FGF2; PetroTech) at 37 °C in a 5% CO_2_ incubator (Forma) [42]. Cell cultures were differentiated in HSC medium containing 0.5% heat-inactivated fetal bovine serum (FBS; Gibco) [45].

### 4.3. Preparation and Treatment with Aβ Peptide

Lyophilized Aβ42 peptide (American Peptide Company, Sunnyvale, CA, USA) was dissolved in hexafluoro-2-propanol (Sigma) to a final concentration of 1 mM. Aliquots of 50 μg were taken, allowed to dry, and stored at −80 °C until use. Oligomeric Aβ42 peptides were prepared by diluting the dry stock in DMSO to 5 mM, then further diluting to 100 µM in Ham’s F-12 Nutrient mix supplemented with GlutaMAX cell culture medium (Thermo Fisher, Waltham, MA, USA; 31765035), and the aliquot was left at 4 °C for 24 h. The next day, the aliquot was diluted to different concentrations for analysis (0.5, 1, and 5 µM) in cell differentiation medium immediately before adding it to cells. hNS1 cells were treated for the first 4.5 days of differentiation. Fibrillary Aβ42 peptides were prepared by diluting the dry stock in DMSO to 5 mM, then further diluting to 100 µM in 10 mM hydrochloric acid solution (prepared in ultrapure H2O from a 1 M HCl stock; Merk; 1.00317.1000), and the aliquot was left at 37 °C for 24 h. The next day, the aliquot was diluted to different concentrations for analysis (0.5, 1, and 5 µM) in cell differentiation medium immediately before it adding to cells. hNS1 cells were treated for the first 4.5 days of differentiation [46]. Untreated cells and vehicle (DMSO)-treated cells were used as controls.

### 4.4. 5′-Bromo-2′-Deoxyuridine (BrdU) Treatment and Detection

To detect proliferating cells, differentiation medium containing 5 µM of 5′-bromo-2′-deoxyuridine (BrdU) (Sigma) was added to the different experimental groups for 2 h. Cell cultures were then immediately washed with PBS and fixed in 4% paraformaldehyde (PFA; Sigma) for 10 min, then washed with PBS again and treated with hydrochloric acid 2 M (HCl; Merck) for 30 min at 37 °C, and finally revealed by immunocytochemistry. BrdU is a thymidine analog and can be incorporated into newly synthesized DNA strands of mitotic cells. The incorporation of BrdU into cellular DNA can then be detected using anti BrdU antibody, allowing assessment of the cell proliferation rate [44].

### 4.5. Immunocytochemistry (ICC)

Cells were rinsed with PBS, fixed in 4% PFA for 10 min, washed with PBS, and blocked for 1 h at room temperature (RT) in PBS containing 0.25% Triton X-100 and 5% normal horse serum (NHS). Primary antibodies were diluted in PBS containing 0.25% Triton X-100 and 1% NHS and incubated overnight at 4 °C. The following antibodies were used: mouse anti-GFAP (1:1000; BD Pharmigen, San Diego, CA, USA), rat anti-BrdU (1:1000; Abcam, Cambridge, UK), rabbit anti-Ki67 (1:500; Thermo Scientific, Waltham, MA, USA), rabbit anti-β-III-tubulin (βIIItub; 1:500; Sigma) and rabbit anti-activated caspase 3 (Casp3; 1:500; Cell Signaling). After removal of primary antibody, cells were washed with PBS and incubated for 1 h at RT with one of the corresponding secondary antibodies: Alexa Fluor 555 donkey anti-mouse IgG, Alexa Fluor 555 goat anti-rat IgG, Alexa Fluor 488 donkey anti-rabbit IgG (1:500; Life Technologies, Grand Island, NY, USA), or Alexa Fluor 488 donkey anti-mouse IgG (1:500; Life Technologies). Finally, nuclei were stained with Höechst 33258 (Hoe; Invitrogen, Waltham, USA) and diluted in PBS (1:1000) for 5 min at RT. Samples were analyzed under a fluorescence microscope (Leica DM IL LED, Wetzlar, Germany). Experiments were repeated three independent times (*n* = 3) with at least three wells per marker for each condition.

### 4.6. RNA Isolation, cDNA Synthesis, and qRT-PCR

Total RNA was isolated with the RNeasy Mini extraction kit (Qiagen, Manchester, UK) according to the manufacturer’s protocol. One microgram of total RNA was reverse transcribed at 50 °C for 60 min in a 20 µL reaction mixture using Super Script III RT (Life Technologies). Relative amounts of cDNA were analyzed by quantitative real-time PCR (qRT-PCR) using the FAST SYBR-green system (Applied Biosystems). Each 15 µL reaction volume included 10 ng total cDNA and 0.3 µM of each primer. qRT-PCRs were performed using primers for the human target genes *GFAP* (forward: 5′-GTTCTTGAGGAAGATCCACGA-3′; reverse: 5′-CTTGGCCACGTCAAGCTC-3′), *TUBB3* (forward: 5′-GCAA CTACGTGGGCGACT-3′; reverse: 5′-ATGGCTCGAGGCACGTACT-3′), and *M**KI67* (forward: 5′-TGACCCTGATGAGAAAGCTCAA-3′; reverse: 5′- CCCTGAGCAACACTGTCTTTT-3′), and the housekeeping gene *TBP* (forward: 5′-GAGCTGTGATGTGA AGTTTCC-3′; reverse: 5′-TCTGGGTTTGATCATTCTGT AG-3′). Amplification of specific PCR products was detected using the SYBR Green PCR Master Mix (Applied Biosystems), according to the manufacturer’s protocol. An Applied Biosystems 7500 Real-Time PCR System was used to determine the amount of target mRNA in each sample, estimated by the 2^−∆∆Ct^ relative quantification method [47]. Gene expression levels were normalized against *TBP* levels in each sample, and the fold change was calculated by setting the expression levels of each gene in the vehicle (DMSO) control as 1.

### 4.7. Western Blot (WB)

To determine the presence of Aβ42 peptide in its oligomeric and fibrillar forms, the differentiation medium with Aβ42 treatment for each condition was collected and analyzed by WB in each experiment. For the detection of cell death, 50 µg of protein extracts were analyzed after Aβ peptide treatment. In both cases, samples were boiled for 5 min, loaded on a 12% sodium dodecyl sulfate (SDS)–polyacrylamide gel, electrophoresed, and transferred to nitrocellulose membranes (GE Healthcare, Little Chal-font, UK). Membranes were blocked in either PBS containing 5% nonfat powdered milk with 0.05% Tween20 (Sigma) or TBS containing 3% BSA and 0.05% Tween20 (to see the phosphorylated state) for 1 h at RT. Blots were incubated overnight at 4 °C with primary antibodies against mouse β-actin (1:1000; Sigma), mouse anti-Aβ 4G8 (1:1000; Covance, Madison, WI, USA) and mouse anti-phospho-Histone H2A.X (γH2AFX; 1:1000; Millipore, Merck). The blots were developed using peroxidase conjugated horse anti-mouse (HAMPO; 1:3000; Vector Laboratories) for 1 h at RT and visualized using the ECL system (Millipore).

### 4.8. Image Analysis and Cell Counting

Analysis and photography of fluorescent cultures were done using a fluorescence microscope (Leica DM IL LED) coupled to a camera (Leica DFC 345 FX). At least eight fields per well were randomly acquired at 40 × magnification to quantify the number of positive cells for the different markers (activated caspase 3, Ki67, BrdU, β-III-tubulin, and GFAP) compared to the total number of cells (Höechst). Each marker was studied in at least three different wells of the same experiment, and each experiment was repeated three independent times (*n* = 3). Cell counting was done using the programs ImageJ (National Institute of Health) and Adobe Photoshop CS6.

### 4.9. Quantification of Pyknotic Nuclei

Apoptotic cells were defined as those exhibiting the morphological hallmarks of apoptosis, such as nuclear fragmentation. At least eight fields per well were randomly acquired at 40x magnification to quantify the number of positive cells for pyknotic nuclei compared to total cells (Höechst). Cell counting was done using the program ImageJ (National Institute of Health).

### 4.10. Statistical Analysis

Statistical analysis was performed using GraphPad Prism 6.0. *p*-values were calculated using one-way ANOVA with post hoc Tukey test. *p*-values < 0.05 were statistically significant (* *p* < 0.05; ** *p* < 0.01; *** *p* < 0.001; ns = not significant). Results are presented as the means ± SD of data from three independent experiments (*n* = 3), with at least three samples per experimental group.

## Figures and Tables

**Figure 1 ijms-22-09537-f001:**
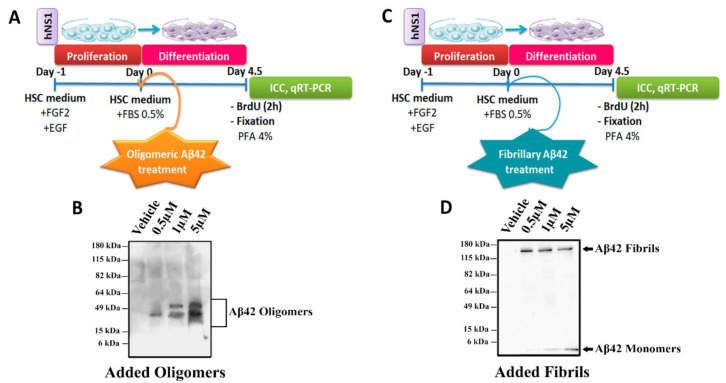
Schematic representation of the experiments and WB analysis. Schematic view of hNS1 cells’ differentiation protocol (See Materials and Methods section) for (**A**) Aβ42 oligomers and (**C**) Aβ42 fibrils. (**B**) Representative WB analysis of Aβ42 forms (using 4G8 antibody) present in extracellular medium used for oligomeric Aβ42 treatment. (**D**) Representative WB analysis of Aβ42 forms (using 4G8 antibody) present in extracellular medium used for fibrillary Aβ42 treatment.

**Figure 2 ijms-22-09537-f002:**
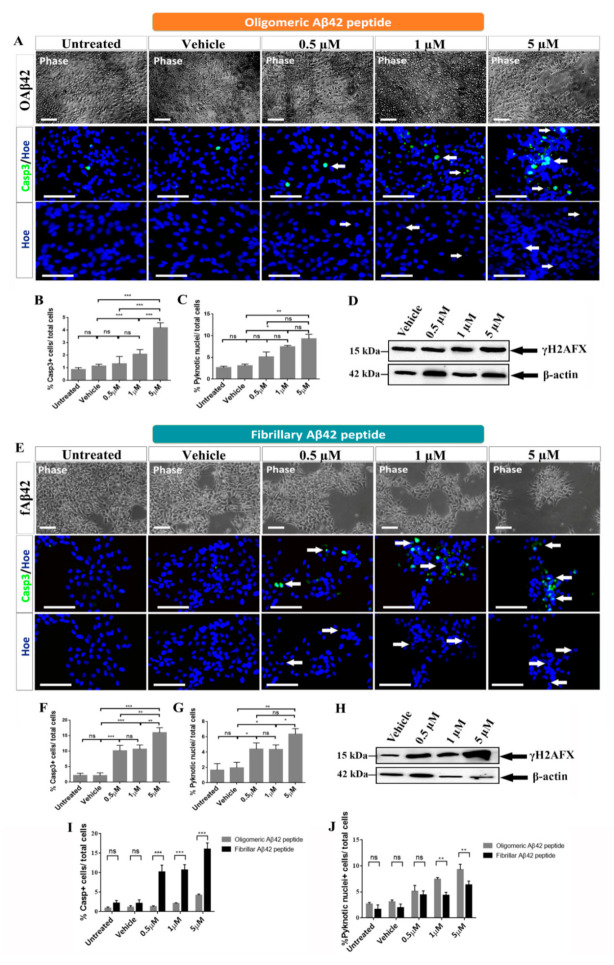
Oligomeric and fibrillary Aβ42 increases cell death in differentiating hNS1 cells. (**A**) Representative phase-contrast images of hNS1 cultures treated with 0.5, 1, and 5 µM of oligomeric Aβ42 peptide (upper panels) and controls (untreated cells and vehicle (DMSO)-treated cells) for 4.5 days. Representative microphotographs of caspase 3 immunoreactivity after oligomeric Aβ42 treatment (middle panels, Casp3, green; see arrows). Representative microphotographs of pyknotic nuclei stained with Höechst are represented in blue (bottom panels, Hoe; see arrows). (**B**) Quantification of the percentage of caspase 3+ cells in response to the specified dose of oligomeric Aβ42 peptide. (**C**) Quantification of the percentage of pyknotic nuclei in the different experimental groups after oligomeric Aβ42 treatment. (**D**) Western blot analysis of double-strand DNA breaks (using γH2AX antibody; 15 kDa) in cellular extracts after oligomeric Aβ42 treatment. (**E**) Representative phase contrast images of hNS1 cultures treated with 0.5, 1, and 5 µM of fibrillary Aβ42 peptide (upper panels) and controls (untreated cells and vehicle (DMSO)-treated cells) for 4.5 days. Representative microphotographs of caspase 3 immunoreactivity after fibrillary Aβ42 treatment (middle panels, Casp3, green; see arrows). Representative microphotographs of pyknotic nuclei stained with Höechst are represented in blue (bottom panels, Hoe; see arrows). (**F**) Quantification of the percentage of Caspase 3+ cells in response to the specified dose of fibrillary Aβ42 peptide. (**G**) Quantification of the percentage of pyknotic nuclei in the different experimental groups after fibrillary Aβ42 treatment. (**H**) Western blot analysis of double-strand DNA breaks (using γH2AX antibody; 15 kDa) in cellular extracts after fibrillary Aβ42 treatment. (**I**) Percentage of caspase 3+/total cells comparing both added forms. (**J**) Percentage of pyknotic nuclei+/total cells comparing both added forms. Scale bar, 100 µM (A, E; upper panels) and 50 µM (A, E; middle and bottom panels). Data are represented as means ± SD of at least three different experiments (*n* = 3). Statistical significance of one-way ANOVA with post hoc Tukey test; * *p* < 0.05; ** *p* < 0.01; *** *p* < 0.001; ns = not significant vs. control groups.

**Figure 3 ijms-22-09537-f003:**
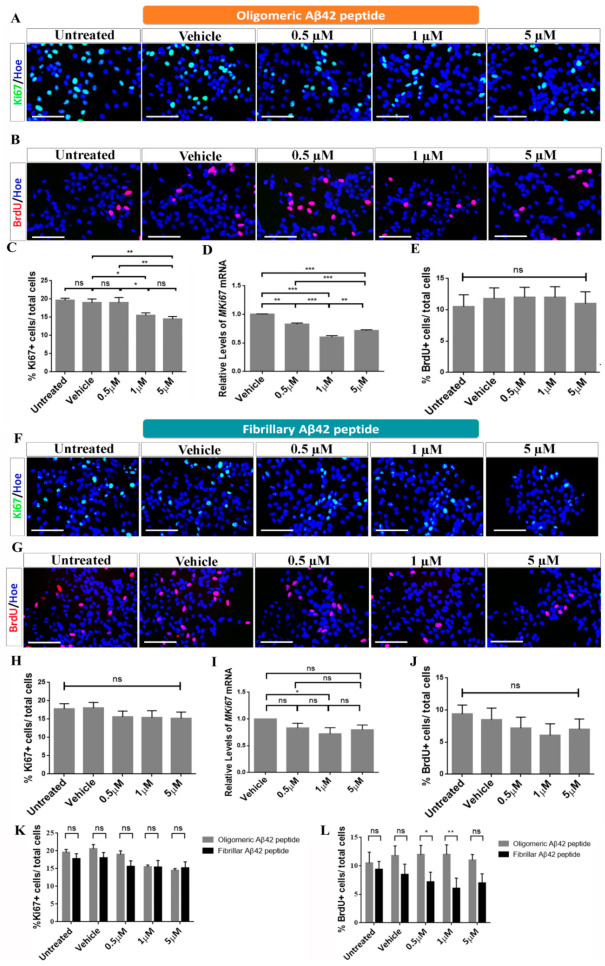
Oligomeric and fibrillary Aβ42 affects cell cycle of differentiating hNS1 cells. (**A**) Representative images showing Ki67 immunoreactivity (green) after oligomeric Aβ42 treatment. (**B**) Microphotographs with immunoreactivity for BrdU (red) after oligomeric Aβ42 treatment. (**C**) Percentage of Ki67+ cells in the different experimental groups after oligomeric Aβ42 treatment. (**D**) Relative expression levels of *M**Ki67* mRNA determined by qRT-PCR analysis after oligomeric Aβ42 treatment. (**E**) Percentage of BrdU+ cells in the different cellular groups after oligomeric Aβ42 treatment. (**F**) Representative images showing Ki67 immunoreactivity (green) after fibrillary Aβ42 treatment. (**G**) Microphotographs with immunoreactivity for BrdU (red) after fibrillary Aβ42 treatment. (**H**) Percentage of Ki67+ cells in the different experimental groups after fibrillary Aβ42 treatment. (**I**) Relative expression levels of *M**Ki67* mRNA determined by qRT-PCR analysis after fibrillary Aβ42 treatment. (**J**) Percentage of BrdU+ cells in the different cellular groups after fibrillary Aβ42 treatment. (**K**) Percentage of Ki67+ cells/total cells comparing both added forms. (**L**) Percentage of BrdU+/total cells comparing both added forms. Nuclei were counterstained in blue with Höechst. Scale bar, 50 µM. Data are represented as means ± SD of at least three different experiments (*n* = 3). Statistical significance of one-way ANOVA with post hoc Tukey test; * *p* < 0.05; ** *p* < 0.01; *** *p* < 0.001; ns = not significant vs control groups.

**Figure 4 ijms-22-09537-f004:**
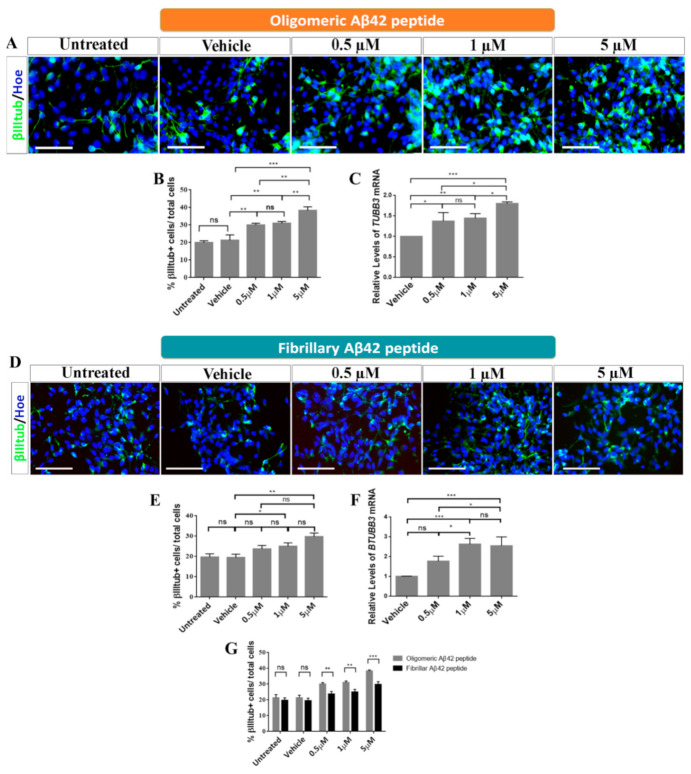
Roles of oligomeric and fibrillary Aβ42 in neurogenesis. (**A**) Representative images showing immunoreactivity for β-III-tubulin (βIIItub; green) after oligomeric Aβ42 treatment. Scale bar, 50 µM. (**B**) Analysis of the percentage of β-III-tubulin+/total cells after oligomeric Aβ42 treatment. (**C**) Relative expression levels of *TUBB3* mRNA by qRT-PCR after oligomeric Aβ42 treatment. (**D**) Representative images showing immunoreactivity for β-III-tubulin (βIIItub; green) after fibrillary Aβ42 treatment. Scale bar, 50 µM. (**E**) Analysis of the percentage of β-III-tubulin+/total cells after fibrillary Aβ42 treatment. (**F**) Relative expression levels of *TUBB3* mRNA by qRT-PCR after fibrillary Aβ42 treatment. (**G**) Percentage of β-III-tubulin+/total cells comparing both added forms. Cell nuclei in A and D were counterstained by Höechst (blue). Data are represented as means ± SD of at least three different experiments (*n* = 3). Statistical significance of one-way ANOVA with post hoc Tukey test; * *p* < 0.05; ** *p* < 0.01; *** *p* < 0.001; ns = not significant.

**Figure 5 ijms-22-09537-f005:**
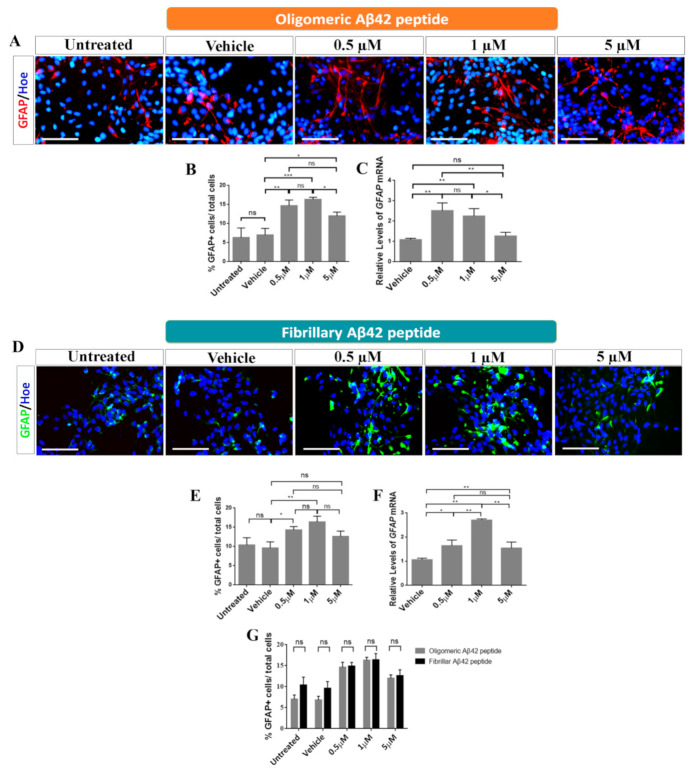
Role of oligomeric and fibrillary Aβ42 in gliogenesis. (**A**) Immunoreactivity for GFAP (red) after oligomeric Aβ42 treatment. Scale bar, 50 µM. (**B**) Analysis of the percentage of GFAP+/total cells in the different groups tested after oligomeric Aβ42 treatment. (**C**) Relative expression levels of *GFAP* mRNA obtained by qRT-PCR after oligomeric Aβ42 treatment. (**D**) Immunoreactivity for GFAP (green) after fibrillary Aβ42 treatment. Scale bar, 50 µM. (**E**) Analysis of the percentage of GFAP+/total cells in the different groups tested after fibrillary Aβ42 treatment. (**F**) Relative expression levels of *GFAP* mRNA obtained by qRT-PCR after fibrillary Aβ42 treatment. (**G**) Percentage of GFAP+/total cells comparing both added forms. Cell nuclei in A and D were counterstained by Höechst (blue). Data are represented as means ± SD of at least three different experiments (*n* = 3). Statistical significance of one-way ANOVA with post hoc Tukey test; * *p* < 0.05; ** *p* < 0.01; *** *p* < 0.001; ns = not significant.

**Figure 6 ijms-22-09537-f006:**
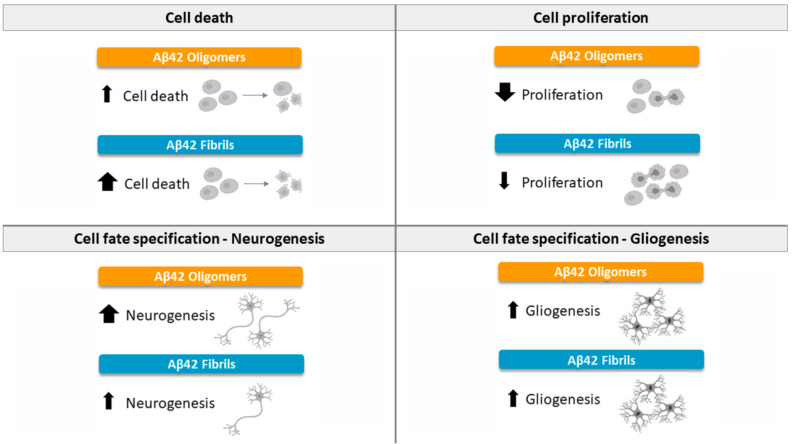
Summary of the effects of Aβ42 peptide (oligomeric and fibrillary) on cell death, proliferation, and differentiation of NSCs. Positive effects are indicated by upward arrows. Negative effects are indicated by downward arrows.

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
