# Peer review of "Oligomeric and Fibrillar Species of Aβ42 Diversely Affect Human Neural Stem Cells"

_ijms, 2021, doi:10.3390/ijms22179537_

Round 1

Reviewer 1 Report

ijms-1359191: Oligomers and fibrillar species of Abeta42 differentially affects human neural stem cell biology

In the present report, the toxic effects of oligomer or fibril Abeta42 on the differentiating human neural stem cells were examined in detail. The fibril Abeta42 caused the apoptotic cell death and damage of DNA stronger than the oligomer Abeta42 did. Only oligomer Abeta42 decreased cell proliferation. The oligomer and Abeta42 caused gliogenesis in a similar way. The oligomer Abeta42 caused neurogenesis more than the fibril Abeta42 did. Since the experiments were performed carefully, this report is worth being published in the Journal. Some minor revisions are suggested as follows, which may help to improve the value of this report.

<Major Points>
(1) Statistical analyses should be performed between the effects of oligomer Abeta42 and those of fibril Abeta42. 
In Fig. 2, 3, 4, and 5, statistical analyses were performed only among oligomer Abeta42 experiments or among fibril Abeta42 ones. The statistical analysis should be performed between oligomer Abeta42 ones and fibril Abeta42 ones. The SD and p value should be indicated in Fig. 6 to show that oligomer Abeta42 and fibril Abeta42 differentially affected hNS1 cells. This statistical analysis may be performed in Fig. 2, 3, 4, and 5 by combining the graphs of each figure, for example, combining Fig. 1B and 1F to one graph.

(2) Statistical analyses should be performed between the effect of a concentration and that of a next lower concentration.
In Fig. 2, 3, 4, and 5, statistical analyses were only performed between the cells with a concentration of Abeta42 and the untreated ones. To evaluate the dose dependency, the p value between each concentration (between 0.5 micromolar and 1.0, between 1.0 and 5.0) should be shown in each graph. In this point of view, some experiments show that 5.0 Abeta42 had less effect than 1.0 Abeta42 (Fig 3 D and I, Fig 5 B, C, E, and F and Fig 6 Gliogenesis). This matter should be discussed in addition to the discussion of line 369. A decrease in the percentage of GFAP+ cells was observed but not the number of them in Fig 5 B and E, which suggested that the reason for the less effect in the highest concentration is not simply the increase of cell death..

(3) The effect of oligomeric Abeta40 and fibril Abeta40
Although there are many subspecies of Abeta, Abeta40 and Abeta42 can be said to be major forms of Abeta as described in the introduction. Although the oligomerization, fibril formation and cellular toxicity of Abeta40 is much less than those of Abeta42, the both forms exist in the brain. Hence, some of the present experiments should be performed using Abeta40. At least, the effect of the species of Abeta other than Abeta42 should be discussed in the context of the use of human neural stem cells for the treatment of Alzheimer’s disease.

(4) The effect of Abeta42 on the cell proliferation.
In the Abstract, line 22, it is described as "The data also show that oligomeric Abeta42 decrease(s) cell(ular) proliferation while Abeta42 fibrils do not affect it". In Fig 3 H, I and J it is true that there was little effect of fibril Abeta42 on the cell proliferation. However, in Fig 6 Cell Proliferation, both oligomer and fibril Abeta42 decrease the cell proliferation. There seems to be a discrepancy.

<Minor Points>
(a) The title " human neural stem cell biology" is not clear. A suggestion is as follows: "Oligomers and fibrillar species of Abeta42 diversely affects differentiating human neural stem cells".

(b) In Fig.1 B and D it is described as "before treatment". The sample of the western blot seems to be just the Abeta peptide added to the culture media. "before treatment" suggests that there may be the data of "after treatment". If the western blot of Abeta42 peptide is performed after treatment (on Day 4.5), the data should be shown and very interesting. If not, simple "added oligomers" and "added fibrils" may be less controversial.

(c) The oligomerization and fibril formation of Abeta42 were performed with surprisingly high efficiency. The reference of the experimental condition should be included (line 440 and line 455).

(d) The differentiation condition needs a reference to show that the differentiation of hNS1 cells proceeded properly (line 444).

(e) The fonts of "cell death" in line 315 and "in the proliferation" in line 324 are bold. The meaning of the usage of bold fonts is unclear. These words may be in normal/plain fonts.

(f) In the abstract, line 26, the full spelling of NSC is necessary.

(g) "Neural Stem Cells (NSCs) ", line 47, should be "neural stem cells (NSCs) ".

(h) In the present study, although the oligomer and fibril Abeta42 increased gliogenesis and cell death, it also promoted neurogenesis. Is there any possibility that Abeta42 is not only neurotoxic but also in some conditions neuroprotective in the brain accumulating Abeta? In the introduction, the experiments using culture cells are explained in detail. A little introduction on the effect of Abeta in the patients' brains may be informative.

End of File

Author Response

Reviewer #1:

Comments and Suggestions for Authors

ijms-1359191: Oligomers and fibrillar species of Abeta42 differentially affects human neural stem cell biology

In the present report, the toxic effects of oligomer or fibril Abeta42 on the differentiating human neural stem cells were examined in detail. The fibril Abeta42 caused the apoptotic cell death and damage of DNA stronger than the oligomer Abeta42 did. Only oligomer Abeta42 decreased cell proliferation. The oligomer and Abeta42 caused gliogenesis in a similar way. The oligomer Abeta42 caused neurogenesis more than the fibril Abeta42 did. Since the experiments were performed carefully, this report is worth being published in the Journal. Some minor revisions are suggested as follows, which may help to improve the value of this report.

We would like to thank the reviewer for the positive comments regarding the manuscript.

<Major Points>
(1) Statistical analyses should be performed between the effects of oligomer Abeta42 and those of fibril Abeta42. 
In Fig. 2, 3, 4, and 5, statistical analyses were performed only among oligomer Abeta42 experiments or among fibril Abeta42 ones. The statistical analysis should be performed between oligomer Abeta42 ones and fibril Abeta42 ones. The SD and p value should be indicated in Fig. 6 to show that oligomer Abeta42 and fibril Abeta42 differentially affected hNS1 cells. This statistical analysis may be performed in Fig. 2, 3, 4, and 5 by combining the graphs of each figure, for example, combining Fig. 1B and 1F to one graph.

We would like to thank the reviewer for the suggestion, which we also considerer interesting. We have added new graphics showing this comparison in the different Figures: Figure 2 (panels I, J), Figure 3 (panels K, L), Figure 4 (panel G) and Figure 5 (panel G). We have also removed the graphics in Figure 6 to avoid confusion.

(2) Statistical analyses should be performed between the effect of a concentration and that of a next lower concentration.
In Fig. 2, 3, 4, and 5, statistical analyses were only performed between the cells with a concentration of Abeta42 and the untreated ones. To evaluate the dose dependency, the p value between each concentration (between 0.5 micromolar and 1.0, between 1.0 and 5.0) should be shown in each graph. In this point of view, some experiments show that 5.0 Abeta42 had less effect than 1.0 Abeta42 (Fig 3 D and I, Fig 5 B, C, E, and F and Fig 6 Gliogenesis). This matter should be discussed in addition to the discussion of line 369. A decrease in the percentage of GFAP+ cells was observed but not the number of them in Fig 5 B and E, which suggested that the reason for the less effect in the highest concentration is not simply the increase of cell death..

As requested by the reviewer, the statistical analysis between all the experimental groups has been carried out, discussed, and added to the graphics in Figures 2, 3, 4 and 5.

(3) The effect of oligomeric Abeta40 and fibril Abeta40
Although there are many subspecies of Abeta, Abeta40 and Abeta42 can be said to be major forms of Abeta as described in the introduction. Although the oligomerization, fibril formation and cellular toxicity of Abeta40 is much less than those of Abeta42, the both forms exist in the brain. Hence, some of the present experiments should be performed using Abeta40. At least, the effect of the species of Abeta other than Abeta42 should be discussed in the context of the use of human neural stem cells for the treatment of Alzheimer’s disease.

We fully agree with the reviewer’s comments. In fact, all these experiments were carried out in parallel with Abeta42 and Abeta40 in the three forms (monomeric, oligomeric and fibrillar) in hNS1 cells. Part of the data from Abeta40 is being collected in another manuscript that will be submitted to this journal in the next few days.

In general, the observed effects for Aβ40 and Aβ42 are quite different. Aβ40 is less toxic to hNS1 cells and favors neurogenesis, with fewer effects on gliogenesis. Aβ42 produces more cell death, favoring gliogenesis, with less effect on neurogenesis. Here, we attach a figure with a summary of the results that we have obtained in our experiments.

(4) The effect of Abeta42 on the cell proliferation.
In the Abstract, line 22, it is described as "The data also show that oligomeric Abeta42 decrease(s) cell(ular) proliferation while Abeta42 fibrils do not affect it". In Fig 3 H, I and J it is true that there was little effect of fibril Abeta42 on the cell proliferation. However, in Fig 6 Cell Proliferation, both oligomer and fibril Abeta42 decrease the cell proliferation. There seems to be a discrepancy.

We apologize about that.  Indeed, as can be observed in Figure 3H (Ki67+ cells) and Figure 3J (BrdU+ cells), in fibrils, there is a trend to decrease proliferation after Abeta42 treatment; but there is not a statistically significant difference between groups.

<Minor Points>
(a) The title " human neural stem cell biology" is not clear. A suggestion is as follows: "Oligomers and fibrillar species of Abeta42 diversely affects differentiating human neural stem cells".

Thank you very much for the suggestion. According to that, the title has been modified in the new version of the manuscript.

(b) In Fig.1 B and D it is described as "before treatment". The sample of the western blot seems to be just the Abeta peptide added to the culture media. "before treatment" suggests that there may be the data of "after treatment". If the western blot of Abeta42 peptide is performed after treatment (on Day 4.5), the data should be shown and very interesting. If not, simple "added oligomers" and "added fibrils" may be less controversial. 

Thank you very much for the comment and suggestion. The Figures have been modified according to the request.

(c) The oligomerization and fibril formation of Abeta42 were performed with surprisingly high efficiency. The reference of the experimental condition should be included (line 440 and line 455).

As requested, we have added a new reference, Nº46 (Bernabeu-Zornoza et al., 2018) to the manuscript, where we describe in detail how the different forms of peptide Aβ can be obtained.

(d) The differentiation condition needs a reference to show that the differentiation of hNS1 cells proceeded properly (line 444).

We totally agree with the reviewer comment. We have been working with hNS1 cells for several years and in different labs. The differentiation protocols and characterization were described in references Nº 42 and 43; additionally, we have included two more recent references (Nº 45 and Nº46) (Coronel et al., 2019; Bernabeu-Zornoza et al., 2018) where hNS1 cells were again characterized and used.

(e) The fonts of "cell death" in line 315 and "in the proliferation" in line 324 are bold. The meaning of the usage of bold fonts is unclear. These words may be in normal/plain fonts.

We apologize for the mistake. We have modified that in the new version of the manuscript.

(f) In the abstract, line 26, the full spelling of NSC is necessary.

Spelling has been added as requested.

(g) "Neural Stem Cells (NSCs) ", line 47, should be "neural stem cells (NSCs) ".

Thank you very much for the appreciation. We have modified that, accordingly.

(h) In the present study, although the oligomer and fibril Abeta42 increased gliogenesis and cell death, it also promoted neurogenesis. Is there any possibility that Abeta42 is not only neurotoxic but also in some conditions neuroprotective in the brain accumulating Abeta? In the introduction, the experiments using culture cells are explained in detail. A little introduction on the effect of Abeta in the patients' brains may be informative.

Yes, we fully agree with the reviewer, in some conditions Aβ42 could be neuroprotective. In fact, the reference Nº12 (Lopez-Toledano and Shelanski., 2004) says that `the effect of Aβ is directly on postmitotic neuronal progenitor cells driving their differentiation toward neurons. Earlier stages of Alzheimer's disease, when the excess of Aβ is enough to form oligomeric but not fibrillar aggregates, the oligomers could activate a compensatory mechanism to replace lost or damaged neurons by increasing the differentiation of neuronal progenitors into new neurons. As time passes and senile plaques are formed, the balance could shift to fibrillar Aβ that could be more neurotoxic. ´ what is in accordance with our results.

Reviewer 2 Report

The authors investigate how oligomer and fibrillar species of Aβ42 affects human neural stem cells differently. I think the results are interesting, but it is unclear to me how the authors' results help to clarify the already contradictory results in the field as discussed in the discussion section.

1) It might be better to state the importance of using hNSC up front in the introduction and not in the discussion. This will help to engage readers who are not in this field as well as justify the reason that the study was conducted in hNSC.

2) As stated in the discussion, it appears that numerous studies have performed similar experiments to study the impact of Aβ42 in hNSC, can the authors make a clear statement on the exact novelty of this study and how this study helps to clarify the contradictory results in the field and not merely add more evidences but also more questions to the field? 

3) How are the Aβ42 oligomer and fibril preparation similar or different from other studies, because this will clear cause discrepancy in the results obtained.

4) For Fig. 1B and 1D, were the WB done with Aβ42 resuspended in the differentiating media, and before treating to the cells? I am a little bit confused by the wording in lines 107-110. Please rephrase if possible.

5) In lines 131-134, it is unclear to me what's the difference between the toxic effect caused by oligomers vs reduced viability caused by fibrils. Are the oligomers and fibrils actually doing the same thing (e.g. being toxic to the cells), or are the authors trying to say that they are having different effects?

6) In lines 166-170, the dose-dependent increase in γH2AFX caused by Aβ42 oligomers is not obvious to me (Fig 2D). But it is obvious in the fibrils. Please clarified this point and perhaps quantified the WB to show the statistical significance.

7) Regarding the GFAP+ staining, the authors suggested that the decrease at 5uM could be due to cell death. However, from the casp3 assay, the cell death was increased from 0.5 to 5uM in a dose dependent manner. Is there other explanation on why the authors observed an increase in the GFAP+ staining for 0.5-1 uM and only a sharp decrease at 5uM?

8) The discussion appears to be a little disorganized and difficult to follow. It will be better to reword this section.

9) I am particular concerned about Figure 6 where the authors basically re-plot their earlier figures (in bar graph) again in a different form (line) to create a summary figure. It's probably not a good idea to do this. I think it's best that the author create a summary figure with propose mechanisms, etc.

Author Response

Reviewer #2.  Comments and Suggestions for Authors

The authors investigate how oligomer and fibrillar species of Aβ42 affects human neural stem cells differently. I think the results are interesting, but it is unclear to me how the authors' results help to clarify the already contradictory results in the field as discussed in the discussion section.

Thank you very much for reviewing the manuscript and for the useful comments and suggestions.

1)It might be better to state the importance of using hNSC up front in the introduction and not in the discussion. This will help to engage readers who are not in this field as well as justify the reason that the study was conducted in hNSC.

We would like to thank the reviewer for the suggestion. Accordingly, we have added a paragraph explaining the importance of hNSCs in the introduction section.

2) As stated in the discussion, it appears that numerous studies have performed similar experiments to study the impact of Aβ42 in hNSC, can the authors make a clear statement on the exact novelty of this study and how this study helps to clarify the contradictory results in the field and not merely add more evidence but also more questions to the field?

Thank you for pointing this out. As commented, there is much controversy about the effects of Aβ peptides, and their different forms, in neural stem cell biology. Probably due, in part, to the fact that each group uses different types of NSC, and that only 1 or 2 different doses of one form of peptide are tested. In this work we have made a systematic analysis of the effects of various doses of Aβ42, in two different forms, using a well characterized line of human NSCs. We are aware that there is still much work to be done.

To emphasize the importance of this work, we have included an improved final paragraph that discusses the relevance and importance of understanding the role of Aβ42 in differentiating NSCs and how this may relate to AD.

3)  How are the Aβ42 oligomer and fibril preparation similar or different from other studies because this will clear cause discrepancy in the results obtained.

The methods to obtain the different forms (monomeric, oligomeric and fibrillar) of the Aβ42 peptides are described in detail in the reference Nº. 46 (Bernabeu-Zornoza et al., 2018), are similar to the used in previous studies by other authors.

4)         For Fig. 1B and 1D, were the WB done with Aβ42 resuspended in the differentiating media, and before treating to the cells? I am a little bit confused by the wording in lines 107-110. Please rephrase if possible.

We apologize for the misunderstanding. We have tried to clarify this aspect in the new paragraph of the manuscript. 

5)         In lines 131-134, it is unclear to me what's the difference between the toxic effect caused by oligomers vs reduced viability caused by fibrils. Are the oligomers and fibrils actually doing the same thing (e.g. being toxic to the cells), or are the authors trying to say that they are having different effects?

Thank you very much for the comment. We have rewritten this paragraph, to make it less confusing, in the new version of manuscript.

6)         In lines 166-170, the dose-dependent increase in γH2AFX caused by Aβ42 oligomers is not obvious to me (Fig 2D). But it is obvious in the fibrils. Please clarified this point and perhaps quantified the WB to show the statistical significance.

We agree with the reviewer; the paragraph (176 to 179) has been rewritten in the new version of the manuscript.

7)         Regarding the GFAP+ staining, the authors suggested that the decrease at 5uM could be due to cell death. However, from the casp3 assay, the cell death was increased from 0.5 to 5uM in a dose dependent manner. Is there other explanation on why the authors observed an increase in the GFAP+ staining for 0.5-1 uM and only a sharp decreases at 5uM?

Thank you very much for the comment. According to our criteria, we believe that the decrease to 5 µM may be related, at least in part, to an increase in cell death, but there could also be a "plateau" effect in the generation of GFAP + cells; that is, maximum differentiation occurs at lower doses of peptide.

8)         The discussion appears to be a little disorganized and difficult to follow. It will be better to reword this section.

We apologize for not explaining this correctly. We have modified some aspects of the discussion in the new version of the manuscript, according to the recommendation of the reviewer.

9) I am particular concerned about Figure 6 where the authors basically re-plot their earlier figures (in bar graph) again in a different form (line) to create a summary figure. It's probably not a good idea to do this. I think it's best that the author creates a summary figure with propose mechanisms, etc.

We apologize for Figure 6; our idea was to make the graphics clearer as a summary of the results, but the effect has not been as desired. In the new version of the manuscript, we have removed the graphics of Figure 6, since throughout all the Figures (2, 3, 4, 5) new graphics have been included comparing the data obtained in oligomers and fibrils in the different experiments (Figure 2 (panels I, J), Figure 3 (panels K, L), Figure 4 (panel G) and Figure 5 (panel G)).
